# Improving transcriptome assembly through error correction of high-throughput sequence reads

Matthew D. MacManes[1] and Michael B. Eisen[1,2,3]

[1] California Institute for Quantitative Biosciences, University of California, Berkeley, CA, USA
[2] Howard Hughes Medical Institute, USA
[3] Department of Molecular and Cell Biology, University of California, Berkeley, CA, USA

## ABSTRACT

The study of functional genomics, particularly in non-model organisms, has been dramatically improved over the last few years by the use of transcriptomes and RNAseq. While these studies are potentially extremely powerful, a computationally intensive procedure, the *de novo* construction of a reference transcriptome must be completed as a prerequisite to further analyses. The accurate reference is critically important as all downstream steps, including estimating transcript abundance are critically dependent on the construction of an accurate reference. Though a substantial amount of research has been done on assembly, only recently have the pre-assembly procedures been studied in detail. Specifically, several stand-alone error correction modules have been reported on and, while they have shown to be effective in reducing errors at the level of sequencing reads, how error correction impacts assembly accuracy is largely unknown. Here, we show via use of a simulated and empiric dataset, that applying error correction to sequencing reads has significant positive effects on assembly accuracy, and should be applied to all datasets. A complete collection of commands which will allow for the production of REPTILE corrected reads is available at https://github.com/macmanes/error_correction/tree/master/scripts and as File S1.

## INTRODUCTION

The popularity of genome enabled biology has increased dramatically, particularly for researchers studying non-model organisms, during the last few years. For many, the primary goal of these works is to better understand the genomic underpinnings of adaptive (*Linnen et al., 2013*; *Narum et al., 2013*) or functional (*Muñoz Merida et al., 2013*; *Hsu et al., 2012*) traits. While extremely promising, the study of functional genomics in non-model organisms typically requires the generation of a reference transcriptome to which comparisons are made. Although compared to genome assembly (*Bradnam et al., in press*; *Earl et al., 2011*), transcriptome assembly is less challenging, significant hurdles still exist (see *Francis et al., 2013*; *Vijay et al., 2013*; *Pyrkosz, Cheng & Brown, 2013* for examples of the types of challenges).

Corresponding author
Matthew D. MacManes,
macmanes@gmail.com

The process of transcriptome assembly is further complicated by the error-prone nature of high-throughput sequencing reads. With regards to Illumina sequencing, error is distributed non-randomly over the length of the read, with the rate of error increasing from 5′ to 3′ end (*Liu et al., 2012*). These errors are overwhelmingly substitution errors (*Yang, Chockalingam & Aluru, 2013*), with the global error rate being between 1% and 3%. While beyond the focus of this paper, the accuracy of *de novo* transcriptome assembly, sequencing errors may have important implications for SNP calling, and the estimation of nucleotide polymorphism and the estimation of transcript abundance.

With regards to assembly, sequencing read error has both technical and 'real-world' importance. Because most transcriptome assemblers use a *de Bruijn* graph representation of sequence connectedness, sequencing error can dramatically increase the size and complexity of the graph, and thus increase both RAM requirements and runtime (*Conway & Bromage, 2011*; *Pell et al., 2012*). More important, however, are their effects on assembly accuracy. Before the current work, sequence assemblers were generally thought to efficiently handle error given sufficient sequence coverage. While this is largely true, sequence error may lead to assembly error at the nucleotide level despite high coverage, and therefore should be corrected, if possible. In addition, there may be technical, biological, or financial reasons why extremely deep coverage may not be possible; therefore, a more general solution is warranted.

While the vast majority of computational genomics research has focused on either assembly (*Chaisson, Pevzner & Tang, 2004*; *Miller, Koren & Sutton, 2010*; *Earl et al., 2011*; *Bradnam et al., in press*) or transcript abundance estimation (*Soneson & Delorenzi, 2013*; *Marioni et al., 2008*; *Mortazavi et al., 2008*; *Pyrkosz, Cheng & Brown, 2013*), up until recently, research regarding the dynamics of pre-assembly procedures has largely been missing. However, error correction has become more popular, with several software packages becoming available for error correction, e.g., ALLPATHSLG error correction (*Gnerre et al., 2011*), QUAKE (*Kelley, Schatz & Salzberg, 2010*), ECHO (*Kao, Chan & Song, 2011*), REPTILE (*Yang, Dorman & Aluru, 2010*), SOAP *denovo* (*Liu, Schmidt & Maskell, 2011*), SGA (*Simpson & Durbin, 2010*) and SEECER (*Le et al., 2013*). While these packages have largely focused on the error correction of genomic reads (with exception to SEECER, which was designed for RNAseq reads), they may likely be used as effectively for RNAseq reads.

Recently a review (*Yang, Chockalingam & Aluru, 2013*) evaluating several of these methods in their ability to correct genomic sequence read error was published. However, the application of these techniques to RNAseq reads, as well as an understanding of how error correction influences accuracy of the *de novo* transcriptome assembly has not been evaluated. Here we aim to evaluate several of the available error correction methods. Though an understanding of the error correction process itself, including its interaction with coverage may be a useful exercise, our initial efforts described here, focus on the the effects of error correction on assembly, the resource which forms the basis of all downstream (e.g., differential expression, SNP calling) steps.

To accomplish this, we simulated 30 million paired-end Illumina reads and assembled uncorrected reads, as well as reads corrected by each of the evaluated correction methods,

which were chosen to represent the breadth of computational techniques used for sequence read error correction. Though we focus on the simulated dataset, we corroborate our findings through the use of an empirically derived Illumina dataset. For both datasets, we evaluate assembly content, number of errors incorporated into the assembly, and mapping efficiency in an attempt to understand the effects of error correction on assembly. Although Illumina is just one of the available high-throughput sequencing technologies currently available, we chose to limit our investigation to this single, most widely used technology, though similar investigations will become necessary as the sequencing technology evolves.

Because the *de novo* assembly is a key resource for all subsequent studies of gene expression and allelic variation, the production of an error-free reference is absolutely critical. Indeed, error in the reference itself will have potential impacts on the results of downstream analyses. These types of error may be particularly problematic in *de novo* assemblies of non-model organisms, where experimental validation of sequence accuracy may be impossible. Though methods for the correction of sequencing reads have been available for the last few years, their adoption has been limited, seemingly because a demonstration of their effects has been lacking. Here, we show that error correction has a large effect on assembly quality, and therefore argue that it should become a routine part of workflow involved in processing Illumina mRNA sequence data. Though this initial work focuses on the results of error correction; arguably the most logical candidate for study, future work will attempt to gain a deeper understanding of error in the error correction process itself.

## RESULTS

Thirty million 100nt paired-end (PE) reads were simulated using the program FLUX SIMULATOR (*Griebel et al., 2012*). Simulated reads were based on the coding portion of the *Mus musculus* genome and included coverage of about 60k transcripts with an average depth of 70X. Thirty million reads were simulated as this corresponds to the sequencing effort suggested by *Francis et al. (2013)* as an appropriate effort, balancing coverage with the accumulation of errors, particularly in non-model animal transcriptomics. These reads were qualitatively similar to several published datasets (*MacManes & Lacey, 2012*; *Chen et al., 2011*). Sequence error was simulated to follow the well-characterized Illumina error profile (Fig. S1). Similarly, the distribution of transcript abundance was typical of many mammalian tissues (Fig. S2), and follows a Poisson distribution with lambda = 1 (*Auer & Doerge, 2011*; *Hu et al., 2011*; *Jiang & Wong, 2009*).

In addition to the simulated dataset, error correction was applied to an empirically derived Illumina dataset. This dataset consists of 50 million 76nt paired-end Illumina sequence reads from *Mus musculus* mRNA, and is available as part of the Trinity software package (*Haas et al., 2013*; *Grabherr et al., 2011*). Because we were interested in comparing the two datasets, we randomly selected 30 million PE reads from the total 50 million reads for analyses. The simulated read dataset is available at http://dx.doi.org/10.5061/dryad.km540, while the empirical dataset may be recreated by subsampling the dataset
**Table 1 Number of raw sequencing reads, sequencing reads corrected, nucleotides (nt) corrected, and approximate runtime for each of the datasets.** Note that neither AllPaths nor Sga provides information regarding the number of reads affected by the correction process.

| Simulated dataset | Total reads | Num reads corr | Num nt corr | Runtime |
|---|---|---|---|---|
| Raw reads | 30M PE | n/a | n/a | n/a |
| AllPathsLG Corr. | 30M PE | ? | 139,592,317 | ~8 h |
| Sga Corr. | 30M PE | ? | 19,826,919 | ~38 min |
| Reptile Corr. | 30M PE | 2,047,088 | 7,782,594 | ~3 h |
| Seecer Corr. | 30M PE | 8,782,350 | 14,033,709 | ~5 h |

available at http://sourceforge.net/projects/trinityrnaseq/files/misc/MouseRNASEQ/mouse_SS_rnaseq.50M.fastqs.tgz/download with the script available at https://github.com/macmanes/error_correction/tree/master/scripts.

Error correction of the simulated and empiric datasets was completed using the Seecer, AllPathsLG, SGA, and Reptile error correction modules. Details regarding the specific numbers of nucleotide changes and the proportion of reads being affected are detailed in Table 1. Despite the fact that each software package attempted to solve the same basic problem, runtime considerations and results were quite different. Trinity assembly using the uncorrected simulated reads produced an assembly consisting of 78.43Mb, while the assembly of empirically derived reads was 74.24Mb.

## Simulated data

Analyses focused on a high-confidence subset of the data, as defined as being 99% similar to the reference over at least 90% of its length. The high-confidence subset of the simulated uncorrected read assembly ($n = 38459$ contigs) contained approximately 54k nucleotide mismatches (Fig. 1), corresponding to an mean error rate of 1.40 mismatches per contig (SD = 7.38, max = 178). There did not appear to be an obvious relationship between gene expression and the quality of the assembled transcripts (Fig. 2). While the rate of error is low, and indeed a testament to the general utility of the *de Bruijn* graph approach for sequence assembly, a dramatic improvement in accuracy would be worth pursuing, if possible.

Error correction of simulated reads using Reptile was a laborious process, with multiple (>5) individual executions of the program required for parameter optimization (specific parameters to be optimized are noted in configuration file https://github.com/macmanes/error_correction/tree/master/scripts, using procedures detailed in README file). While each individual run was relatively quick, the total time exceed 12 h, with manual intervention and decision making required at each execution. Error correction resulted in the correction of 7.8M nucleotides (of a total ~5B nucleotides contained in the sequencing read dataset). The resultant assembly contains an average of 1.23 mismatches per contig (SD = 6.46, max = 152). The absolute number of errors decreased by ~12% (Fig. 1), which represents substantial improvement, particularly given that the high
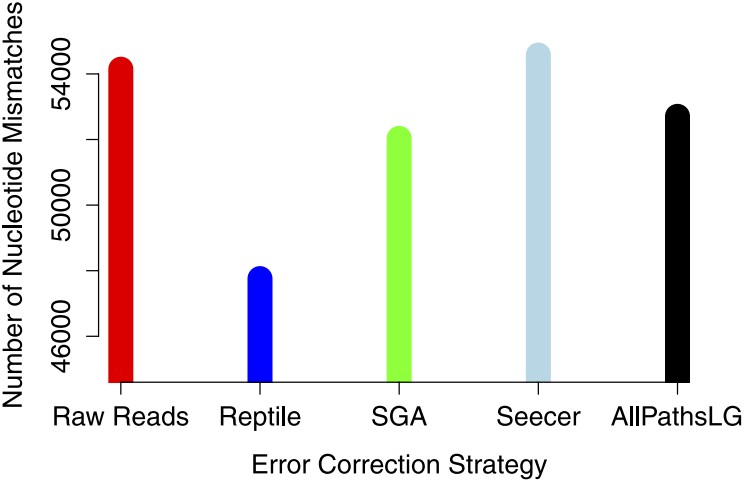

**Figure 1** **The global estimate of nucleotide mismatch decreases with error correction.** The assembly done with Reptile corrected reads has approximately 10% fewer errors than does the raw read assembly.

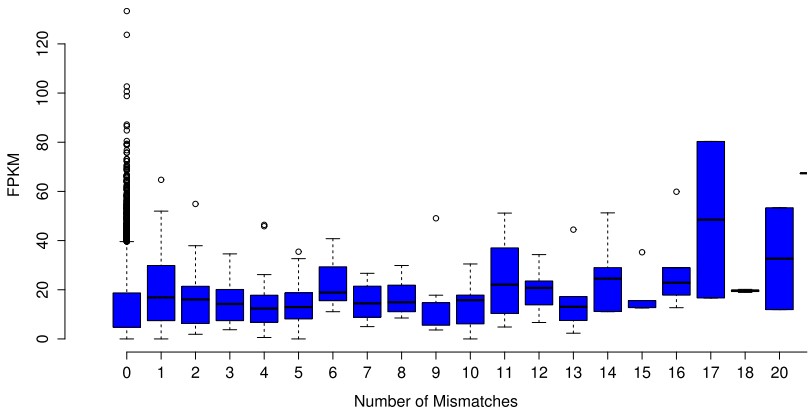

**Figure 2** **The number of nucleotide mismatches in a given contig is not related to gene expression.** On average, in the assembly of uncorrected simulated reads, poorly expressed transcripts are no more error prone than are highly expressed transcripts.

confidence subset of the Reptile-corrected assembly was the largest ($n = 38670$ contigs) of any of the methods (Table 2).

AllPathsLG error correction software implemented by far the most aggressive correction, selected optimized parameters in an automated fashion, and did so within a 4 h runtime. AllPathsLG corrected nearly 140M nucleotides (again, out of a total ~5B nucleotides contained in the sequencing reads), which resulted in a final assembly with 52706 nucleotide errors, corresponding to a decrease in error of approximately 2.7%.

Seecer, is the only dedicated error-correction software package dedicated to RNAseq reads. Though Seecer is expected to handle RNAseq datasets better than the other correction programs, its results were disappointing. More than 14 million nucleotides were changed, affecting approximately 8.8M sequencing reads. Upon assembly 54,574

**Table 2 Assembly details.** High confidence datasets included only contigs that matched a single reference, had sequence similarity >99%, and covered ≥90% of length of reference.

| Dataset | Error corr. method | Raw assembly size | High conf. size |
|---|---|---|---|
| Simulated reads | | | |
| | None | 64491 (78Mb) | 38459 (27Mb) |
| | AllPathsLG | 64682 (78Mb) | 38628 (27Mb) |
| | Sga | 65059 (80Mb) | 38619 (27Mb) |
| | Reptile | 63099 (73Mb) | 38670 (25Mb) |
| | Seecer | 65468 (80Mb) | 38407 (27Mb) |
| Empiric reads | | | |
| | None | 57338 (74Mb) | 21406 (24Mb) |
| | AllPathsLG | 53884 (66Mb) | 21204 (23Mb) |
| | Sga | 56707 (75Mb) | 21323 (24Mb) |
| | Reptile | 53780 (60Mb) | 21850 (22Mb) |
| | Seecer | 57311 (75Mb) | 21268 (24Mb) |

nucleotide errors remained which is equivalent to the number of errors contained in the assembly of uncorrected reads.

Lastly, Sga error correction was implemented on the simulated read dataset. Sga is the fastest of all error correction modules and finished correcting the simulated dataset in 38 min. The software applied corrections to 19.8M nucleotides. Its correction resulted in a modest improvement in error, with a reduction in error of approximately 4% over the assembly of uncorrected errors.

Assembly content, aside from fine-scaled differences at the nucleotide level, as described above, were equivalent. Assemblies consisted of between 63,099 (Reptile) – 65,468 (Seecer) putative transcripts greater than 200nt in length. N50 ranged from 2319 (Reptile) – 2403nt (Sga). The high-confidence portion of the assemblies ranged in size from 38407 contigs (Seecer assembly) to 38670 contigs in the Reptile assembly. Assemblies are detailed in Table 2, and available at http://dx.doi.org/10.6084/m9.figshare.725715.

The proportion of reads mapping to each assembled dataset was equivalent as well, ranging from 92.44% using raw reads to 94.89% in Sga corrected reads. Assemblies did not appear to differ in general patterns of contiguity (Fig. 3) though it should be noted that the most successful error corrector, Reptile had both the smallest assembly size *and* largest number of high confidence contigs. Taken together, these patterns suggest that error correction may have a significant effect on the structure of assembly; though its major effects are in enhancing resolution at the level of the nucleotide. Indeed, while we did not find, nor expect to find large differences in these global metrics, we do expect to see a significant effect on transcriptome based studies of marker development and population genetics, which are endeavors fundamentally linked to polymorphism, estimates of which can easily be confused by sequence error.

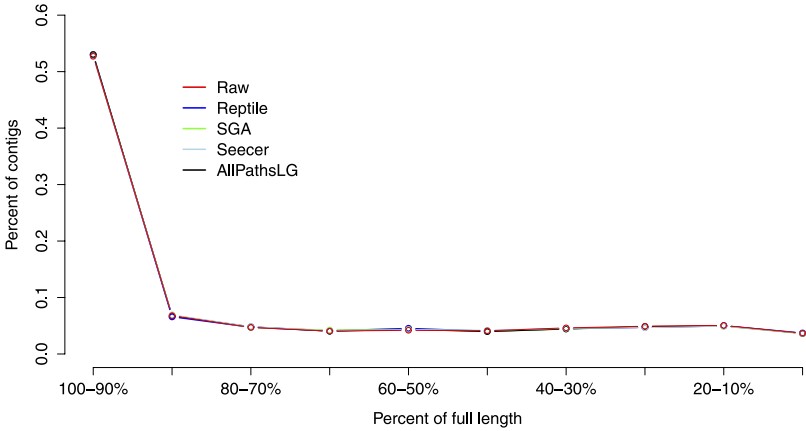

**Figure 3 Assembly contiguity did not vary significantly between assemblies of reads using the different error correction methods.** Each error correction methods, as well as assembly of raw reads, produced an assembly that is dominated by full length (both start and stop codon present) or nearly full length assembled transcripts.

## Empirical data

The high-confidence subset of the uncorrected empirical read assembly ($n = 21406$ contigs) contained approximately 14.7k nucleotide mismatches, corresponding to a mean error rate of .68 mismatches per contig (SD = 3.60 max = 197). Error correction procedures were implemented as described above. Indeed, the resultant pattens of correction were recapitulated. Error correction using REPTILE were most favorable, and resulted in a reduction in the number of nucleotide errors by more than 10%, to approximately 13k. As above, the high-confidence portion of the REPTILE-corrected dataset was the largest, with 21580 contigs, which is slightly larger that the assembly of uncorrected reads. Similar to what was observed in the simulated dataset, the high-confidence portion of the ALLPATHS corrected assembly was the smallest of any of the datasets, and contained the most error. Of interest, the SGA correction performed well, similar to as in simulated reads, decreasing error by more than 9%.

Empirical assemblies contained between 53780 (REPTILE) and 57338 (uncorrected assembly) contigs greater than 200nt in length. N50 ranged from between 2412 (REPTILE) and 2666nt (SEECER) in length. As above, assemblies did not differ widely in their general content or structure; instead effects were limited to differences at nucleotide level. Assemblies are available at http://dx.doi.org/10.6084/m9.figshare.725715.

## DISCUSSION

Though the methods for error correction have become increasingly popular within the last few years, their adoption in general genome or transcriptome assembly pipelines has lagged. One potential reason for this lag has been that their effects on assembly, particularly in RNAseq, has not been demonstrated. Here, we attempt to evaluate the effects of four different error correction algorithms on assembly, arguably the step upon which all downstream steps (e.g., differential expression, functional genomics, SNP discovery, etc.) is based. We use both simulated and empirically derived data to show a significant effect of

correction on assembly, especially when using the error corrector REPTILE. This particular method, while relatively labor intensive to implement, reduces error by more than 10%, and results in a larger high-confidence subset relative to other methods. Aside from a reduction in the total number of errors, REPTILE correction both reduced variation in nucleotide error, and reduced the maximum number of errors in a single contig.

Interesting, SEECER, the only error correction method designed for RNAseq reads, performed relatively poorly. In simulated reads, SEECER slightly increased the number of errors in the assembly, though with applied to empirically derived reads, results were more favorable, decreasing error by ~3%. Though the effects of coverage on correction efficiency were not explored in the manuscript describing SEECER (*Le et al., 2013*), their empirical dataset contained nearly 90 million sequencing reads, a size $3\times$ larger than the dataset we analyze here. Future work investigating the effects of coverage on error correction is necessary.

In addition to this, how error correction interacts with the more complicated reconstructions, splice variants for instance, is an outstanding question. Indeed, reads traversing a splicing junction may be particularly problematic for error correctors, as coverage on opposite sides of the junction may be different owing to differences in isoform expression, which could masquerade as error. Alternative splicing is known to negatively affect both assembly and mapping (*Vijay et al., 2013*; *Sammeth, 2009*; *Pyrkosz, Cheng & Brown, 2013*), and given that many computational strategies are shared between these techniques and error correction suggests that similarly, error correction should be affected by splicing. Indeed, many of the most error-rich contigs were those where multiple isoforms were present. As such, considering this potential source of error in error correction should be considered during error correction. Computational strategies that distinguish these alternative splicing events from real error are currently being developed.

The effects of read coverage on the efficiency of error correction are likely strong. Aside from the suggestion that SEECER's relatively poor performance owed to low coverage data relative to the dataset tested during the development of that software (*Le et al., 2013*), other supporting evidence exists. Approximately 5% of reads are miscorrected. When looking at a sample ($n = 50000$) of these reads, the contig to which that read maps is on average more lowly expressed than appropriately corrected reads (Fig. 4, Wilcoxon rank sum test, $W = 574733$, $p$-value $= 0.00022$), which suggests that low coverage may reduce the efficiency of error correction. In addition, miscorrected reads, whose average expression is lower, tend to have more corrections than to the appropriately corrected reads (Fig. 5, $t$ test, $t = -2.1755$, df $= 7164.8$, $p$-value $= 0.029$).

Though sequence read error correction failed to have a large effect on global assembly metrics, there was substantial improvement at the nucleotide level. Indeed, these more fine scaled effects are both harder to assay, particularly in non-model organisms, and also potentially more damaging. For instance, one popular application for transcriptome assembly is for population genomics. Most population genomics analysis are fundamentally based on estimates of polymorphism, and higher polymorphism, stemming from error, may bias results in unpredictable ways. In addition to error's effects on estimation

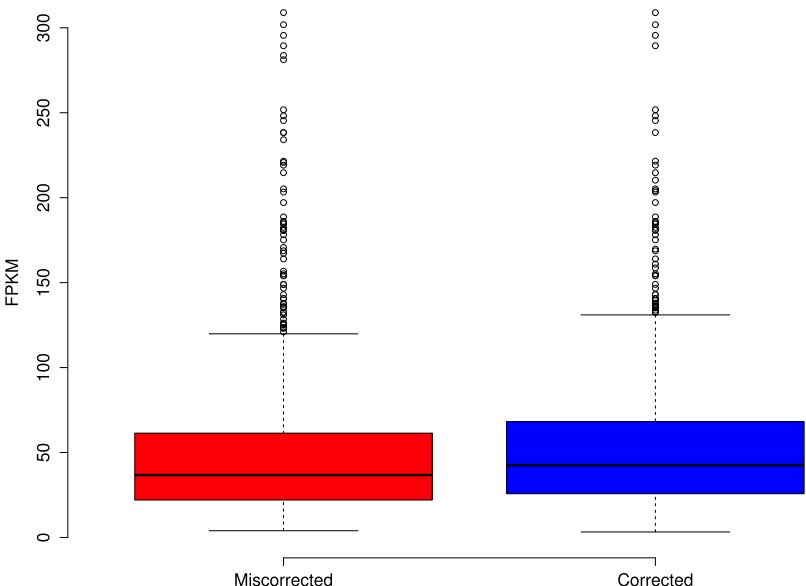

**Figure 4 Reads miscorrected by Reptile have lower expression, on average, than to appropriately corrected reads.**

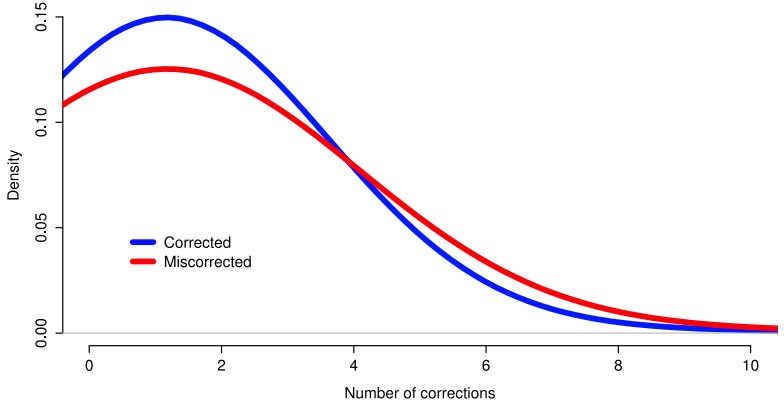

**Figure 5 Reads miscorrected by Reptile have more corrections, on average, than to appropriately corrected reads.**

of polymorphism, researchers interested in studying functional biology may also be impacted. Here, insertion errors may create nonsensical amino acid translation of a coding sequence, while more common substitution errors may form premature stop codons. Though errors remain even after error correction, a reduction in magnitude of error is certainly something worth pursuing.

Finally, software for the correction of short read data appear to be becoming more popular, with a recent review (*Yang, Chockalingam & Aluru, 2013*) detailing several. Though not exhaustive, we attempted to assay at least a single software package implementing each of the existing computational methods for error correction. While we intended to assay a corrector implementing the MSA (multiple sequence alignment) method, ЕСНО (*Kao, Chan & Song, 2011*), this was not possible. Though ЕСНО was promising, execution

was prohibitively slow, and its execution was killed after ∼336 h. A similar issue was encountered in the *Yang, Chockalingam & Aluru (2013)* review. Moving forward, there appears to be a need for additional RNAseq-specific error correctors, especially when sequencing depth is moderate to low. Work currently in progress aims to accomplish in this in an automated, memory-efficient fashion.

## METHODS

Because we were interested in understanding the effects of error correction on the assembly of vertebrate transcriptome assembly, we elected to use coding sequences greater than 200nt in length from the *Mus musculus* reference genome (GRCm37.71), available at http://uswest.ensembl.org/Mus_musculus/Info/Index. Thirty million 100nt paired-end Illumina reads were simulated with the program FLUX SIMULATOR (*Griebel et al., 2012*) which attempts to simulate a realistic Illumina RNAseq dataset, incorporating biases related to library construction and sequencing. Thirty million PE reads were simulated as this sequencing effort was suggested to be optimal for studies of whole-animal non-model transcriptomes (*Francis et al., 2013*). Sequencing error increased along the length of the read, as per program default. Patterns of gene expression were modeled to follow patterns typically seen in studies of Eukaryotic gene expression. The FLUX SIMULATOR requires the use of a parameter file, which is available at https://github.com/macmanes/error_correction/tree/master/scripts.

In addition to analyses conducted on a simulated dataset, we used the well-characterized mouse dataset included with the Trinity software package (http://sourceforge.net/projects/trinityrnaseq/files/misc/MouseRNASEQ/mouse_SS_rnaseq.50M.fastqs.tgz/download) to validate the observed patterns using an empirically derived dataset. To enable comparison between the simulated and empiric dataset, we randomly selected a subset of this dataset consisting of 30 million PE reads.

Quality metrics for simulated and experimental raw reads were generated using the program SOLEXAQA (*Cox, Peterson & Biggs, 2010*), and visualized using R (*R Core Team, 2012*). Patterns of gene expression were validated using the software packages BOWTIE2 (*Trapnell et al., 2010*) and EXPRESS (*Roberts & Pachter, 2013*). All computational work was performed on a 16-core 36GB RAM Linux Ubuntu workstation.

Error correction was performed on both simulated and empirical datasets using four different error correction software packages. These included SEECER, ALLPATHSLG error correction, REPTILE, and SGA. These specific methods were chosen in an attempt to cover the breadth of analytical methods currently used for error correction. Indeed, each of these programs implements a different computational strategy for error correction, and therefore their success, and ultimate effects on assembly accuracy are expected to vary. In addition, several of these packages have been included in a recent review of error correction methods, with one of these (REPTILE) having been shown to be amongst the most accurate (*Yang, Chockalingam & Aluru, 2013*).

Though error correction has been a part of the ALLPATHSLG genome assembler for the past several versions, only recently has a stand-along version of their Python-based

error correction module (http://www.broadinstitute.org/software/allpaths-lg/blog/?p=577 and *Maccallum et al., 2009*), which leverages several of the AllPaths subroutines, become available. With exception to the minimum kmer frequency, which was set to 0 (unique kmers retained in the final corrected dataset), the AllPathsLG error correction software was run using default settings for correcting errors contained within the raw sequencing reads. Code for running the program is available at https://github.com/macmanes/error_correction/tree/master/scripts.

Error correction using the software package Reptile requires the optimization of several parameters via an included set of scripts, and therefore several runs of the program. To correct errors contained within the raw dataset, we set kmer size to 24 (*KmerLen = 24*), and the maximum error rate to 2% (*MaxErrRare = 0.02*). Kmer = 24 was selected to closely match the kmer size used by the assembler Trinity. We empirically determined optimal values for *T_expGoodCnt* and *T_card* using multiple independent program executions. Reptile requires the use of a parameter file, which is available at https://github.com/macmanes/error_correction/tree/master/scripts.

The software package SGA was also used to correct simulated and empiric Illumina reads. This program, like AllPaths-LG, allows its error correction module to be applied independent of the rest of the pipeline. These preliminary steps, preprocessing, indexing, and error correction were run with default settings, with exception to the kmer size, which was set to 25.

Lastly, the software package Seecer was used to error correct the raw read dataset. The software package is fundamentally different than the other packages, in that it was designed for with RNAseq reads in mind. We ran Seecer using default settings.

Transcriptome assemblies were generated using the default settings of the program Trinity (*Haas et al., 2013*; *Grabherr et al., 2011*). Code for running Trinity is available at https://github.com/macmanes/error_correction/tree/master/scripts. Assemblies were evaluated using a variety of different metrics. First, Blast+ (*Camacho et al., 2009*) was used to match assembled transcripts to their reference. TransDecoder (http://transdecoder.sourceforge.net/) was used to identify full-length transcripts. For analysis of nucleotide mismatch, we elected to analyze a 'high-confidence' portion of out dataset as multiple hits and low quality BLAT matches could significantly bias results. To subset the data, we chose to include only contigs whose identity was ≥99% similar to, and covering at least 90% of the reference sequence. The program Blat (*Kent, 2002*) was used to identify and count nucleotide mismatches between reconstructed transcripts in the high-confidence datasets and their corresponding reference. Differences were visualized using the program R.

## CONCLUSIONS

To evaluate the effects of correction of sequencing error on assembly accuracy, we generated a simulated Illumina dataset, which consisted of 30M paired-end reads. In addition, we applied the selected error correction strategy to an empirically derived *Mus musculus* dataset. We attempted error correction using four popular error correction

software packages, and evaluated their effect on assembly. Though originally developed with genome sequencing in mind, we found that all tested methods do correct mRNAseq reads, and increase assembly accuracy, though REPTILE appeared to have the most favorable effect. This study demonstrates the utility of error correction, and proposes that it become a routine step in the processing of Illumina sequence data.

## ACKNOWLEDGEMENTS

This paper was greatly improved by suggestions from members of the Eisen Lab, and from two named reviewers, C. Titus Brown and Mick Watson.

### Funding

MDM is supported by a NIH NRSA Postdoctoral fellowship (1F32DK093227). MBE is a Howard Hughes Medical Institute Investigator. The funders had no role in study design, data collection and analysis, decision to publish, or preparation of the manuscript.

### Grant Disclosures

The following grant information was disclosed by the authors:
NIH NIDDK: 1F32DK093227.

### Competing Interests

The authors declare no competing interests exist.

### Author Contributions

- Matthew D. MacManes conceived and designed the experiments, performed the experiments, analyzed the data, wrote the paper.
- Michael B. Eisen wrote the paper.

### Data Deposition

The following information was supplied regarding the deposition of related data:
Dryad: http://dx.doi.org/10.5061/dryad.km540
Figshare: http://dx.doi.org/10.6084/m9.figshare.725715
GitHub: https://github.com/macmanes/error_correction/tree/master/scripts.

### Supplemental Information

Supplemental information for this article can be found online at http://dx.doi.org/10.7717/peerj.113.

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
