# Peer review of "Improving transcriptome assembly through error correction of high-throughput sequence reads"

_PeerJ, doi:10.7717/peerj.113_

## Round 0.1 · original submission · Minor Revisions

From my initial review I did see the utility of the error-correction practice, but the conclusions section for me fell a bit short. If you really want to sell this methodology I would like to suggest an outline of proposed steps to be taken pointers to how they differ from other error-correction methods.

If you choose to strengthen the impact that this manuscript might have, I do agree with the comments in the use of 'real' vs 'fake' data. There are numerous datasets available from the NCBI SRA resource that might suffice for 'real' data.If the reader would like to duplicate your tests, where would the test dataset be available from? If you are considering FigShare for this purpose, some additional pointers to the data should be included. This would allow the readers to identify with your methods being able to test the suggestions on a local system before trusting the application on ones own data.

The methodology does appear a bit out-of-date focusing on only one commercial system and being trained on single-end reads. Some software offerings when applied to transcriptome data attempt prediction and accounting for alternate splicing. An error-reducing approach may also need to deal with building perhaps a statistical significance to splice variation. Since this manuscript mainly focused on the Trinity software, a similar treatment should be applied to the other assemblers mentioned. In formulating a recommended pipeline, using various data preparation and assemblers, perhaps a table showing the pipeline and contrasting the differences/similarities would greatly familiarize the reader with the steps to take in handling transcriptome data. With some genomes shown to be quite complex, would you also recommend some of these methods for non-transcriptome data? Other simple features such as using set up-tools such as 'jellyfish' may also be useful; it's not for error correction, but is a fast k-mer assessor tool and may be useful in data preparation. Again since the reader will be using 'real' data, setting the stage for which tools would be needed are quite useful to the readers.

I know the reviewers suggested 'minor revisions'; however, I feel that if you reflect on the comments here and in the earlier arXiv submission, you may have a stronger message to present here. I would feel that this would require a 'moderate revision'; the option is yours, but you must realize that the technologies in this area are moving rapidly, and you have a chance to meet the current trends. The reviewers did have some valid points that should be addressed and their evaluations should help guide your re-submission. Thank you for submitting to PeerJ and I hope to see the updated version.

Below are some specific suggestions to ease reading of the manuscript:

Example of annotation:
LINE NO.: /PREVIOUS FORM/SUGGESTED FORM/ [ADDITIONAL NOTE]

abstract: /// [try using more formal punctuations such as a comma, period, semi-colon instead of the '–'.]
abstract: /// [depending on punctuation changes, something must be adjusted between 'reference transcriptome' and 'must be completed'.]
line 8: /–- transcriptome assembly is less challenging, significant/; transcriptome assembly is less challenging and significant/[re-word or use more formal punctuation.]
line 21: / de bruijn / de Bruijn /[]
line 23: /// [one reviewer suggested adding pertinent citations which would strengthen the section. Options suggested were: for the impact of errors, Conway
& Bromage (2011); Pell et al. (PNAS 2012); or for digital normalization (Brown et al, arXiv).]
line 23: / importantly, / important; /[]
line 47: /, assembled uncorrected reads, /and assembled uncorrected reads, /[]
line 48: /, and /, /[remove 'and']
line 122: /-- instead, it's /; instead its /[adjusted punctuation.]
line 158: /subroutines, become available. /subroutines./[]
line 166: /I set kmer size to 25/the kmer size was set to 25/[]
line 168: /I empirically determined optimal values/The optimal values were empirically determined/[]
line 173: /developed be a group/developed by a group/[]
line 176: /I split the dataset/the dataset was split/[]
Figure 2: /\textsc//[This may need correction, I didn't see what was referred to here.]

·

Basic reporting

This is good.

Experimental design

This is good.

Validity of the findings

Data is not available. See general comments for rest.

Additional comments

This is an interesting paper examining the effect of running several
short-read error-correction software packages -- most targeted at
Illumina reads, from genomes -- on mRNAseq data.

I already got a chance to comment on this back when it was released on
arXiv, so this is some ways a second review (see the Haldane's Sieve
commentary)! The authors addressed some of my comments on their
earlier draft, but not all. Plus, I read this version of the
paper more thoroughly.

At an overall level, this paper suffers from two fairly significant
defects. First, the authors did not apply their approaches to real
data, despite a surfeit of real data in the real world. Thus their
results are of somewhat unknown relevance to real data, which
(unfortunately) differs from fake data -- I've written a whole paper
on this in metagenomics! Second, I am concerned that the analysis of
the error correction is not very deep, as it is largely confined to
looking at the end stage of things -- assembled contigs -- after
applying error correction at the front-end. I would be very
interested in seeing an analysis of the error correction output
on the raw reads; for example, how many reads were uncorrected? how
many reads were miscorrected? etc.

To some extent these two defects are tradeoffs: it's a lot harder to
know when reads are miscorrected in real data. But I think one of
them needs to be addressed before publication. Alternatively, I would
be amenable to a strongly worded statement saying something like "this
is only an initial investigation of the whole pipeline, because (a) if
it didn't work on fake data, why try on real data? and (b) nobody
cares about anything but the final assembly anyway." If you wanted
to try out real data, the Trinity mouse data set is pretty nice and
easily accessible.

(A third defect is that their data was not made available to
reviewers. What?!)

Some more specific comments that should be addressed in words, if not
in deed:

1) mRNAseq has variable coverage, which should throw off coverage
estimates critical to most error correction algorithms. I would expect
to see both miscorrected reads and uncorrected reads resulting from
this. Do you see high coverage reads that were uncorrected or
miscorrected, and do you see low coverage reads that were corrected?

2) Splice variants. I would expect error correction to mess with
splice junctions where you have a high-coverage exon joined to both
another high-coverage exon OR a low-coverage exon, because those
should look like errors. I haven't thought about how to analyze this
so it would be a real contribution if you could figure that out for me

3) Assemblers already apply error correction internally (bubble
removal, etc.) It would be nice to hear speculation why Fig 2 looks
like it does -- is it simply that assemblers suck at error correction?
* * *
Minor comments:

abstract: "must be completed" should probably have a -- in front of it.

line 21, "de bruijn" => "de Bruijn" (or at least double check)

on line 23, I think a good citation for the impact of errors is Conway
& Bromage (2011), although I'm not sure they described actual numbers
for a real assembler. For that, you might look at Pell et al. (PNAS
2012), or the digital normalization paper (Brown et al, arXiv),
although it has to be inferred from some of the tables.

line 23, "importantly" => "important"

line 47, "assembled reads" => "and assembled reads"

line 48, first "and" is probably unnecessary.

line 122, "it's" => "its"

line 173, "be" => "by"

line 176, "I" => "we"?

Figure 2, "\textsc" comes through in caption.

·

Basic reporting

No comments

Experimental design

No comments

Validity of the findings

No Comments

Additional comments

MacManes et al describe the effects of four read correction tools on transcriptome assembly (performed by Trinity), concluding that there are large variations in single nucleotide resolution, but that the final results of the assemblies are largely unchanged (i.e. contiguity is not affected by error correction).

The research is well described and is well suited to publication in PeerJ.

However, I feel it could be improved in a number of ways:

- the paper focuses only on Illumina data. A large of body of research exists on assembly of 454 data (e.g. http://www.plosone.org/article/info%3Adoi%2F10.1371%2Fjournal.pone.0051188 and http://www.biomedcentral.com/1471-2164/11/571). With Ion Proton threatening to challenge Illumina, it is possible that a comparison across technologies would be very interesting. I acknowledge that this is a lot of extra work.

- the choice of error correction tools appears arbitrary. SOAPdenovo is mentioned but not used. SGA (which carries out two different types of error correction) is neither mentioned nor used. The choice of tools should be justified.

- why was single end data used? Most transcriptome assembly is carried out with paired end data as this should improve assembly. Also why 50M reads? MiSeq produces 15M, HiSeq 2500 150M, HiSeq 2000 180M and GAIIx 30M. The choice of 50M should be justified.

- would using % error figures look better in figure 2?

- A discussion of why errors end up in the assembly could be added. Why do errors end up in the assembly? If errors are in the minority compared to correct reads, why doesn't coverage solve the problem? With errors, do we end up with two versions of the transcript, one containing error, the other not?

- What is distribution of error across length of transcripts i.e. are errors enriched at the ends of transcripts i.e. do errors lead to transcript breaks?

- Some discussion on the consequences of ignoring single nucleotide errors could also be given e.g. if you correct, do you end up with more correct ORFs? If you don't correct, do you end up with more erroneous stop codons leading to erroneous pseudogene prediction etc

Mick Watson

---

## Round 0.2 · Minor Revisions

You incorporated well the suggestions of the reviewers and your points toward incorporating error-correction methods prior to assembly is well received. This provides a nice outline of what has been the practiced norm for methods in sequence assembly and demonstrates the improvements in assembly quality that can be gained with key attention to sequence quality through error correction. The expanded explanation of the tools used and resources for the tools in the Methods section greatly enhance the utility of the manuscript. I would consider this version ACCEPTED except for some adjustments needed for the web resource and some remaining notes. We're almost there. Great work!

QUAKE, ECHO, and SOAPdenovo were mentioned as containing error correction algorithms, but not necessarily tested. A statement regarding any knowledge toward where these software may stand among the mix that were tested here would also be beneficial. A note of the lengthy time requirement for ECHO would even be useful.

As mentioned, the dropbox.org resources will need to moved to a resource such as Dryad or Figshare. The PeerJ resource reserves 50Mb for supplemental materials if needed. Since the Figshare resource was used for some of the assembly data perhaps the personal account github scripts, a suggested NOTE for line 116, and a README file describing the contents would be better pooled together and made accessible as a single *.tar.gz file. Everything in one place would be even better.

Suggested edits:

Abstract:
I guess I'm just not a fan for the use of the dashes; is it that important? If you are going to use them, then use them sparingly and correctly; you already have three in the manuscript. The correct usage as I see it would as follows:

The study of functional genomics -–particularly in non-model organisms-- has been dramatically improved over the last few years by use of transcriptomes and RNAseq.

While these studies are potentially extremely powerful, a computationally intensive procedure –-the de novo construction of a reference transcriptome-- must be completed as a prerequisite to further analyses.

If you disagree, please point me to a punctuation guide that approves of this usage. I don't think I'm interpreting the clear break as you intended it.

For the rest of manuscript:

Example of annotation:
LINE NO.: /PREVIOUS FORM/SUGGESTED FORM/ [ADDITIONAL NOTE]

ABSTRACT: /list of commands with will/list of commands which will/ [change word]

ABSTRACT: /https://gist.github.com/macmanes/5878728 // [personal account]

line 23: /assemblers were thought/assemblers were generally thought/ [since not all assemblers were considered a global assumption cannot be made here]

line 27: /be possible, therefore, a more/be possible; therefore, a more/ [add semicolon]

line 78/79: /Supplementary Figure 1/Figure 1/ [The figure is part of the main manuscript]

line 79: /patterns of gene expression/patterns related to gene expression/ [Can this last sentence be re-worded to better explain what you mean. Also see line 147.]

line 80: /Supplementary Figure 2/Figure 2/ [The figure is part of the main manuscript]

line 88: /https://www.dropbox.com/s/mp8fu0tijox69ki/simulated.reads.tar.gz// [ANOTHER REPOSITORY NEEDED]

line 89: /https://www.dropbox.com/s/rkl0ihqom28smb2/empiric.reads.tar.gz// [ANOTHER REPOSITORY NEEDED]

line 111: /observe an obvious/obvious/ [mal-aligned duplication needs fixing]

line 116: /was a laborious process/was a laborious process (see NOTE)/ [Someone trying to do this would be helped understanding what conditions need to be met to adjust the parameters]

line 147: //Gene expression is represented as Fragments Per Kilobase of transcript per Million mapped reads (FPKM)/ [EXPLAIN Y AXIS]

line 157/158: /http://dx.doi.org/10.6084/m9.figshare.725715 // [ACCEPTABLE REPOSITORY]

line 192/193: /http://dx.doi.org/10.6084/m9.figshare.725715 // [ACCEPTABLE REPOSITORY]

line 209: /// [is the dash really needed here?]

line 254: /assembly is population genomics/assembly is for population genomics/ [for the topic area of rather than the topic itself]

line 264: /http://uswest.ensembl.org/Mus_musculus/Info/Index // [ACCEPTABLE REPOSITORY]

line 272: /https://gist.github.com/macmanes/5859902 // [personal account]

line 275: /http://sourceforge.net/projects/trinityrnaseq/files/misc/MouseRNASEQ/mouse_SS_rnaseq.50M.fastqs.tgz/download // [ACCEPTABLE]

line 297: /http://www.broadinstitute.org/software/allpaths-lg/blog/?p=577 // [NOT SURE THAT THIS BLOG TOPIC WILL BE MAINTAINED]

line 301: /https://gist.github.com/macmanes/5859931 // [personal account]

line 309: /https://gist.github.com/macmanes/5859947// [personal account]

line 321/322: /https://gist.github.com/macmanes/5859956 // [personal account]

line 324: /http://transdecoder.sourceforge.net// [OK as project data source]

line 445: /http://arxiv.org/abs/1303.2411v1ar// [OK as reference]

END OF EDITS

---

## Round 0.3 · accepted · Accept

Thank you for the quick turn-around on the suggested edits. I believe you have done everything possible to help make the manuscript read well and have provided the best possible access to the supplementary tools and data. I hope that this will set a precedent in treating the raw data beforehand to attain the best possible assemblies. I myself will be trying to optimize local datasets based on your recommendations and see if we can improve upon what has already been done.

It looks like everything asked for is in place; the supplementary files do appear to be in order. There is a possiblity that a few extra refinements may be needed and I'm sure the PeerJ staff will work with you to make sure everything is satisfactory.